# The ratio of expansion to compression: A new measure of lifespan disparity

**Zhen Zhang**[1], **Qiang Li**[2]*

**1** Institute of Population Research, School of Social Development and Public Policy, Fudan University, Shanghai, P. R. China, **2** Population Research Institute, School of Social Development, East China Normal University, Shanghai, P. R. China

* qli@soci.ecnu.edu.cn

## Abstract

Numerous studies have shown that high life expectancy is closely related to low life disparity. Unlike life expectancy, which can be increased by mortality decline at any age, life disparity can either increase or decrease in response to mortality decline. Disparity can thus be decomposed into two opposite components, called compression and expansion, depending on the effect of mortality decline on the age distribution of mortality. Without specifying the two components, various conventional measures of disparity may provide misleading information relating to how life chances in society can be equalized. Based on the relevant properties of changes in disparity, we develop a new measure of disparity—the ratio of expansion to compression—that can account for the relative importance of the two components. This simple measure not only provides a clear view of the evolution of disparity, but also permits changes in disparity related to mortality decline to be interpreted in a consistent manner similar to life expectancy. Simulations and an empirical analysis demonstrated the advantages of this new measure over conventional measures of disparity.

## Introduction

As the average lifespan of humans has increased [1], how such gains in longevity are distributed among the members of society has received increased attention [2, 3]. Many studies have shown a strong negative correlation between life expectancy and lifespan disparity regardless of what measures of disparity are used, including life years lost due to death ($e^{\dagger}$) [4, 5], Gini coefficient [6], standard deviation in age at death [7], and life table entropy (i.e., the ratio of $e^{\dagger}$ to life expectancy at birth) [8–10]. Over the last 160 years, humans have experienced both increases in lifespan and an equalization of life chances [11].

Previous studies have shown that the properties of lifespan disparity are different from those of life expectancy [12]. Mortality decline at any age increases life expectancy. However, averting deaths may either increase or reduce life disparity. The sign of the effect depends on the age at which deaths are averted. For a population having a life table with entropy lower than one, there is a threshold age such that averting deaths before that age reduces disparity but averting deaths after that age increases disparity.

Accordingly, we can decompose lifespan into two parts: one before the threshold age and the other after. Based on the concept of mortality compression, we refer to the parts before and after

**Data Availability Statement:** The Human Mortality Database is freely available at www.mortality.org.

**Funding:** This work was supported by the National Natural Science Foundation of China to ZZ (Grant No. 71473044) and Shanghai Planning Office of Philosophy and Social Science to QL

(2019BSH002) and to ZZ (2020BSH014). The funders had no role in study design, data collection and analysis, decision to publish, or preparation of the manuscript.

**Competing interests:** The authors have declared that no competing interests exist.

the threshold age as the compression component and the expansion component, respectively, because mortality decline before the threshold age compresses mortality whereas that after the threshold age leads to expansion. With mortality decline, the two components change in opposite directions; thus, the observed change in disparity represents a balance between the two parts.

Because of the mechanism underlying changes in disparity, an increase in lifespan disparity is not always unfavorable. For instance, an increase in lifespan disparity (here indicated by lost life because of death; see below) may correspond to two different stories. In a country with increasing younger adult mortality (because of wars or political distress), the compression component can rise and increase lifespan disparity. In contrast, in another country that has experienced much progress in reducing old-age mortality, which often coincides with a decline in younger adult mortality, the age distribution of deaths is stretched out to older ages. The mortality expansion component increases and outweighs mortality compression, leading to an increase in the overall disparity. Thus, stories can show striking differences despite the same amount of change in lost life. In the former case, the increase in lifespan provides information on the deterioration in lifespan disparity, whereas the increase in lifespan of the latter case is positive because more people are surviving to older ages. Thus, an aggregate measure of lifespan disparity itself cannot indicate precisely how equal life chances are in a society.

Although detailed examination of the age-specific change in lifespan disparity is sufficient for characterizing the actual implications of the aggregate measure, a single measure related to the change in lifespan would provide useful information on the dynamics of lifespan and disparity.

Here, we developed a new measure that combines the two opposite components into a single indicator. Although there are various ways that this measure could be developed in theory, we consider a simple, straightforward measure: the ratio of the expansion component to the compression component. We first explore this new measure with simulations and then apply it to empirical data to compare it with conventional measures. Generally, this new measure provides new insight into the evolution of lifespan disparity.

## Materials and methods

### Mechanisms behind changes in life disparity

Life disparity can be measured by life expectancy lost due to death [9, 12, 13]

$$e^{\dagger} = \int_{0}^{\omega} d(a)e(a)da,$$

where $e(a) = \int_{a}^{\omega} \ell(x)dx/\ell(a)$ is the remaining life expectancy at age $a$, $d(a) = \ell(a)\mu(a)$ is the life table distribution of deaths, and $\omega$ is the maximum lifespan to reach. Note that $\ell(a) = \exp\{-\int_{0}^{a} \mu(x)dx\}$ indicates the probability of survival to age $a$ with $\ell(0) = 1$, where $\mu(a)$ denotes the age-specific hazard of death.

When a population has a life table with entropy less than one (as in most populations), there exists a unique age, denoted by $a^{\dagger}$, such that lifesaving (mortality decline) before this age decreases lifespan disparity and lifesaving after this age increases lifespan disparity. At the threshold age, the lifespan disparity can thus be decomposed into two parts—a compression component and an expansion component—in terms of the direction of the impact of mortality decline on lifespan disparity:

$$e^{\dagger} = \underbrace{\int_{0}^{a^{\dagger}} d(a)e(a)da}_{\text{Compression}} + \underbrace{\int_{a^{\dagger}}^{\omega} d(a)e(a)da}_{\text{Expansion}} = e_{c}^{\dagger} + e_{e}^{\dagger} \tag{1}$$

Although the expansion and the compression parts are separately defined, they are intimately associated with each other. First, they add up to the overall lifespan disparity, and a change in either of them results in a change in the other. A change in mortality at any age can change many functions of the life table, including the age distribution of deaths and the remaining life expectancy at all ages. These functions are the most commonly used building blocks of many disparity measures, such as variance in age at death, the Gini coefficient index [14], as well as life lost due to death. The disparity either before or after the threshold age can be affected by a change in mortality, as shown by a previous perturbation analysis [15].

Second, and more importantly, any change in mortality can change the threshold age by which the overall disparity is decomposed into expansion and compression parts. As shown in Eq (1), the threshold age determines the age interval over which the expansion and comparison components are calculated, and shifts in the threshold age to younger or older ages extend or shrink the corresponding age intervals, leading to changes in both components. Hence, the two components are closely linked.

Furthermore, relative changes—not absolute changes—are most important for interpreting changes in disparity in the two components. In principle, the threshold age is independent of life expectancy because the mean of a random variable is irrelevant to the statistics of dispersion (e.g., variance or other similar measures). However, in practice, the threshold age of lifespan disparity ($e^\dagger$) is located near the age of life expectancy and is generally 2 years or so lower than the life expectancy from the 1950s onward [12]. In other words, the threshold age for most populations is old age, meaning that the compression component has a much larger age interval than the expansion component. The compression component is thus usually greater than the expansion component and is more likely to experience larger absolute changes. Because the baseline scales are different, comparisons of the absolute change in the two components often make little sense. Therefore, a measure that reflects the importance of one component relative to another is needed to understand lifespan disparity.

## The ratio of expansion to compression

To obtain a comprehensive but simplistic view of changes in lifespan, we combine the two pieces of information into a single measure: the ratio of the expansion component to the compression component (REC):

$$REC = \frac{Expansion}{Compression} = \frac{e^\dagger_e}{e^\dagger_c}. \tag{2}$$

The change in REC is essentially the change in the relative importance of the expansion component to the compression component, which is related to changes in $d(a)$, $e(a)$ and $a^\dagger$ caused by changes in mortality as show in Eq (1).

The REC can increase in two cases. The first case is a rise in young-age mortality, which increases the compression component. Moreover, the increase in young-age mortality lowers the threshold age and widens the interval $[a^\dagger, \omega]$, thereby further augmenting the expansion component. When the expansion term rises by more than the compression term, the REC increases.

In the second case, when old-age mortality declines faster than young-age mortality, the expansion component can increase more than the compression component decreases, or the expansion component decreases less than the compression component decreases, causing the REC to rise. Note that steady improvement in old-age mortality can increase the threshold age. The REC may decrease if the threshold age is high and, thus, leaves the compression

component with a much larger interval $[0, a^\dagger]$ than that for the expansion component $[a^\dagger, \omega]$. This decrease stems from the expansion term's increasing less than the compression decreases for any given decrease in mortality (see below). The increase in the expansion term can outweigh the increase in the compression term if there is sufficiently great progress of mortality improvement among older people. Additionally, if life expectancy approaches a limit (a topic still under debate [16, 17]), the interval $[a^\dagger, \omega]$ contracts, thereby leading to a decrease in the expansion component. The ultimate outcomes, thus, depend on the amount of remaining room for further improvement in human mortality, particularly at advanced ages.

Similarly, the REC can decrease in two cases. In the first case, when death rates among younger adults decline at a faster pace than among older people, the interval $[0, a^\dagger]$ will widen, and the resulting increase in the compression component can exceed that in the expansion component. Consequently, the REC will decrease. In the second case, the expansion component decreases due to a fast rise in threshold age, which squeezes the interval $[a^\dagger, \omega]$, accompanied by the relatively slow pace of old-age mortality reduction, while the compression component may decrease slowly or even increase; thus, the REC decreases.

Analogous to the REC, a single indicator can be obtained by subtracting the expansion component from the compression component, which yields the difference between expansion and compression (DEC):

$$DEC = Expression - Compression = e_e^\dagger - e_c^\dagger. \tag{3}$$

The change in DEC reveals that a positive DEC reflects the dominance of compression, and a decline implies that lifespan disparity decreases (and vice versa). The DEC ranges from negative to positive values. In principle, both REC and DEC can work equally well for measuring the change or difference in lifespan disparity. In this study, we mainly focus on the REC because the DEC can be positive or negative and requires extra care when interpreting negative values.

## Decomposition of the change over time of the REC

The change over time in life disparity is given by [12]

$$\dot{e}^\dagger(t) = \int_0^\omega \rho(x, t)g(x, t)dx, \tag{4}$$

where $\rho(x, t) = -\dot{\mu}(x, t)/\mu(x, t)$ represents the age-specific rates of mortality decline over time, $\dot{\mu}(x, t) = d\mu(x, t)/dt$. The function $g(x, t) = d(x, t)[e^\dagger(x, t) + e(x, t)(H(x, t) - 1)]$, where $H(x, t) = \int_0^x \mu(a, t)da$ is the cumulative hazard function, represents the change of $e^\dagger$ caused by a reduction in mortality at age $a$, where $e^\dagger(x, t) = \int_x^\omega d(a, t)e(a, t)da/\ell(x, t)$ is the life disparity above age $x$. Thus, $g(x,t)$ indicates how much $e^\dagger$ will change given a decrease in mortality at age $x$, that is, the efficiency of mortality reductions in changing $e^\dagger$.

The change in $e^\dagger$ can be broken into two parts at the threshold age:

$$\dot{e}^\dagger(t) = \int_0^{a^\dagger} \rho(x, t)g(x, t)dx + \int_{a^\dagger}^\omega \rho(x, t)g(x, t)dx = \dot{e}_c^\dagger(t) + \dot{e}_e^\dagger(t). \tag{5}$$

The first term on the right-hand side of (5) represents the change in the compression component and the second the change in the expansion component. The balance of the two components determines whether the lifespan disparity increases or decreases. Hence, if $e^\dagger$ is time constant, then

$$\dot{e}^\dagger(t) = 0 \Longleftrightarrow \dot{e}_e^\dagger(t) = -\dot{e}_c^\dagger(t) \ or \ \frac{\dot{e}_e^\dagger(t)}{\dot{e}_c^\dagger(t)} = -1. \tag{6}$$

The disparity will rise when $\dot{e}_e^\dagger$ is greater than $-\dot{e}_c^\dagger$ and vice versa.

It follows from Eqs (2) and (4) that

$$\frac{d \ln REC(t)}{dt} = \frac{d \ln e_e^\dagger(t)}{dt} - \frac{d \ln e_c^\dagger(t)}{dt} = \frac{\dot{e}_e^\dagger(t)}{e_e^\dagger(t)} - \frac{\dot{e}_c^\dagger(t)}{e_c^\dagger(t)}, \tag{7}$$

which means that the relative change of the REC is determined by the difference in the relative change between expansion and compression. The REC can increase if the expansion component increases or the compression component decreases (i.e., $\dot{e}_c^\dagger / e_c^\dagger < 0$). Consequently, the REC can generate consistent conclusions relating to survival status in terms of disparity.

Similar to (6), we have the following relation:

$$\frac{d \ln REC(t)}{dt} = 0 \Longleftrightarrow \frac{\dot{e}_e^\dagger(t)}{e_e^\dagger(t)} = \frac{\dot{e}_c^\dagger(t)}{e_c^\dagger(t)} \Longleftrightarrow \frac{\dot{e}_e^\dagger(t)}{\dot{e}_c^\dagger(t)} = \frac{e_e^\dagger(t)}{e_c^\dagger(t)}. \tag{8}$$

The REC can remain unchanged when the pace of the relative change of the expansion component is the same as that of the compression component, or equivalently, when the ratio of the relative changes in the expansion and compression terms is equal to the ratio of the two terms themselves—that is, the REC itself. If the relative change in the expansion term is greater than that in the compression term, the REC increases (and vice versa).

Comparison of Eqs (6) and (8) reveals that the REC is more sensitive than $e^\dagger$ in response to mortality improvement. It follows from Eq (8) that the REC can rise if $\dot{e}_e^\dagger$ is greater by a factor of $e_e^\dagger / e_c^\dagger$ than $\dot{e}_c^\dagger$. In contrast, $e^\dagger$ can rise only when $\dot{e}_e^\dagger$ is greater than $-\dot{e}_c^\dagger$. In other words, a relatively small change in the expansion component can be reflected by the REC, whereas a perceptible rise in $e^\dagger$ requires a large change in the expansion component. The reason behind this difference lies in that the REC is a relative measurement, but $e^\dagger$ is an absolute measurement. Because $e_e^\dagger$ is much smaller than $e_c^\dagger$, it is difficult for the increase in $e_e^\dagger$ to exceed the change in $e_c^\dagger$ in terms of absolute values. However, because $e_e^\dagger$ is much smaller than $e_c^\dagger$, its relative change is more likely to be greater than the relative change of $e_c^\dagger$. In the sense, the REC is more appropriate than $e^\dagger$ for measuring lifespan disparity.

## Results

### Illustrative example

To demonstrate how the ratio works, we begin with an illustrative example of a rise in $e^\dagger$ for Russian men and Japanese women in the period of 1990–1995, compared to a decrease in $e^\dagger$ for Belgian men in 1960–1965 and England & Wales men in 2000–2005, based on data obtained from the Human Mortality Database (www.mortality.org) [18].

Fig 1 depicts the age trajectories of $g(x)$ and $\rho(x)$ for four selected country-based populations. In Fig 1A, the points at which the function $g(x)$ crosses the horizontal line are the threshold ages, marked by the vertical lines in corresponding colors and line types. Decreased mortality before the threshold age can yield a greater change, precisely a decline, in $e^\dagger$ compared with the possible change for decreases in mortality after the threshold age. Fig 1B displays the relative rate of progress against mortality at different ages, $\rho(x)$. Despite the universal pattern of mortality decline in the developed countries [19], the selected populations show a pronounced difference in the mortality improvement's age pattern. The changes in $e^\dagger$, $e_c^\dagger$ and $e_e^\dagger$ depend on the product of $\rho(x)$ and $g(x)$, as indicated in Eqs (4) and (5).

Mortality in Russia deteriorated in the 1990s because of social distress [20], as shown in Fig 1B, and the life expectancy of Russian men fell from 63.76 to 58.12 years, coupled with an increase in life disparity from 14.70 to 15.68 years (Table 1). The distribution of deaths shifted leftward, with a surge of deaths at middle age (Fig 2A). In the same period, Japanese women

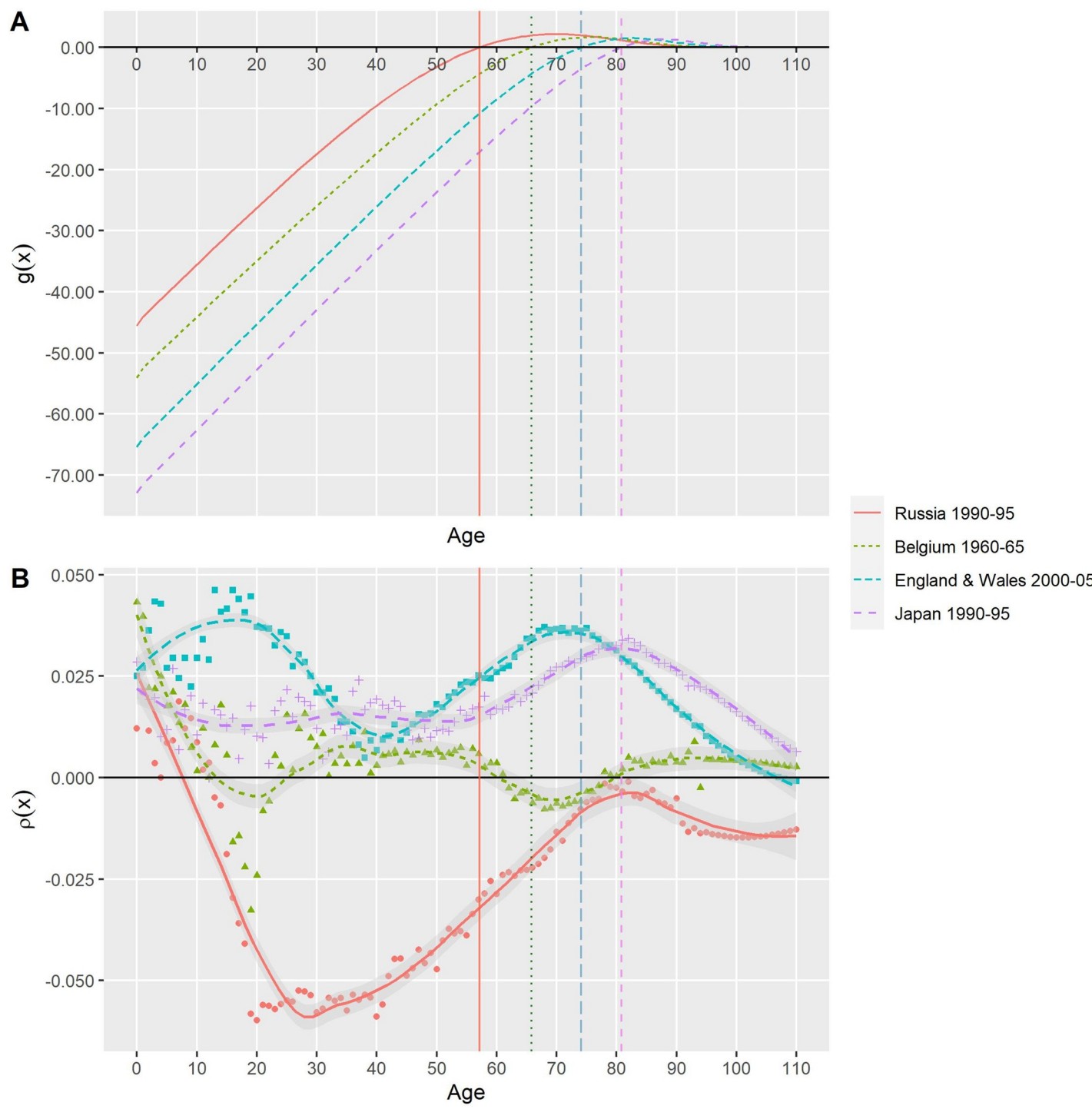

**Fig 1. Age trajectories of $g(x)$ and $\rho(x)$, selected country-based populations. Source: Human Mortality Database (2020).**

experienced increases in life expectancy, particularly at older ages (Fig 1B); more of them survived to and died at advanced ages. The distribution of deaths resultantly shifted rightward (Fig 2B). Despite mortality improvement, the life disparity of Japanese women increased from 9.29 to 9.42 years. Based on this increase, it is challenging to distinguish whether each case

Table 1. Life expectancy, life disparity and the REC, selected country-based populations.

| | Russian men | | Japanese women | | Belgian men | | England & Wales men | |
|---|---|---|---|---|---|---|---|---|
| Year | 1990 | 1995 | 1990 | 1995 | 1960 | 1965 | 2000 | 2005 |
| $e^o$ | 63.76 | 58.12 | 81.87 | 82.78 | 66.71 | 67.56 | 75.62 | 77.17 |
| $e^\dagger$ | 14.70 | 15.86 | 9.29 | 9.42 | 13.55 | 13.12 | 10.80 | 10.59 |
| $a^\dagger$ | 60.87 | 53.08 | 80.4 | 81.36 | 65.49 | 65.60 | 73.34 | 75.29 |
| $e_c^\dagger$ | 9.64 | 9.82 | 6.08 | 6.20 | 9.31 | 8.70 | 6.93 | 6.97 |
| $e_e^\dagger$ | 5.06 | 6.04 | 3.21 | 3.22 | 4.25 | 4.41 | 3.87 | 3.63 |
| REC | 0.53 | 0.62 | 0.53 | 0.52 | 0.46 | 0.51 | 0.56 | 0.52 |

represents survival deterioration or improvement. The REC, however, provides clarification: it rose for Russian men, indicating the increasing dominance of expansion, and decreased for Japanese women, indicating the decreasing dominance of compression.

The same $e^\dagger$ decline may result from different compositions of expansion and compression. In 1960–1965, $e^\dagger$ for Belgian men fell from 13.55 to 13.12 years. The compression term decreased but the expansion term increased, leading the REC rise from 0.46 to 0.51. Despite the similar decline of $e^\dagger$ for Belgians, men in England and Wales had the opposite situation: the compression component increased, while the expansion decreased, yielding an REC decrease from 0.56 to 0.52. The difference in the REC change can be attributed to somewhat different age distributions of the progress of reducing mortality. In Belgium, male youths underwent increased mortality, which can decrease $e_c^\dagger$, when the threshold age continued to rise owing to mortality improvement at other ages. The progress of declining old age thus became increasingly important in the change of $e^\dagger$, as evidenced by an increase in $e_e^\dagger$. Males in England and Wales enjoyed mortality improvement, particularly among boys and older men (Fig 1B). The progress at older ages, however, is not great enough to offset the impact of the rise in threshold age and the resulting increase in $e_c^\dagger$; therefore, the REC decreased.

Table 2 presents the decomposition of the relative changes in the REC. Russian men experienced increases in both expansion and compression, but expansion increased more rapidly than compression did; consequently, REC increased. Japanese women also experienced increases in both components, but unlike Russian men, the rate of change of expansion was slower than that of compression; as a result, REC decreased. The results of decomposition show that, even with the similar decline of $e^\dagger$, Belgian men and England and Wales men had the opposite composition of changes in $e_c^\dagger$ and $e_e^\dagger$

## Simulations

To demonstrate the impact of the age distribution of mortality improvement on changes in REC, we considered three different scenarios of age-based patterns of mortality decline. The first was the case of shifting mortality, wherein we assumed that the rate of mortality decline was the same for all ages. The second was "faster progress above threshold age $a^\dagger$," in which the death rates above the threshold age $a^\dagger$ declined faster than those below; the third was "faster progress below threshold age $a^\dagger$."

We conducted a "classical" simulation based on a Siler model with a changing force of mortality as in [21, 22]:

$$\mu(x, t) = \alpha_1 e^{-\rho_1 t - \beta_1 x} + \alpha_2 e^{-\rho_2 t} + \alpha_3 e^{-\rho_3 t + \beta_3 x}, \qquad (9)$$

where the parameters $\beta_1$ and $\beta_3$ are the rate of infant mortality declining with age and the rate

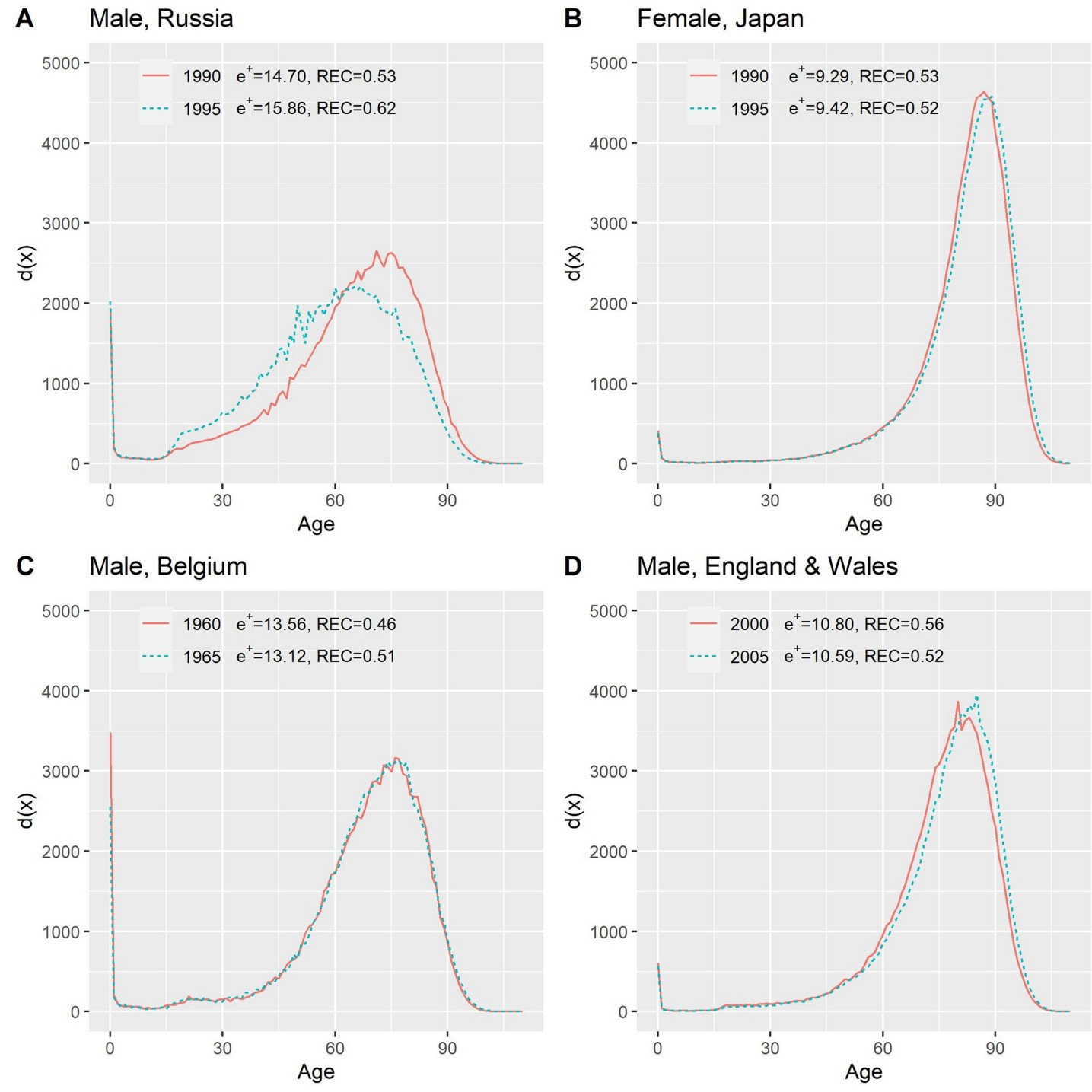

**Fig 2. Changes the age distribution of deaths with changes in life disparity, selected country-based populations.**

of adult mortality increasing with age, respectively; $\mu(0,0) = \alpha_1+\alpha_2+\alpha_3$ is the initial mortality when $t = 0$ and $x = 0$. The values of these parameters were set as $\alpha_1 = 1\times10^{-9}$, $\alpha_2 = 3\times10^{-4}$, $\alpha_3 = 1\times10^{-6}$, $\beta_1 = 1\times10^{4}$, and $\beta_3 = 0.13$, which were approximated based on the period mortality rates of Japanese females in 2017. The parameters $\rho_1$ and $\rho_2$ are set constants over time (both

**Table 2. Decomposition of relative change in REC, selected country-based populations.**

|  | Russian men | Japanese women | Belgian men | England & Wales men |
|---|---|---|---|---|
|  | 1990–1995 | 1990–1995 | 1960–1965 | 2000–2005 |
| Observed ln$REC/dt$ | 0.032 | -0.003 | 0.021 | -0.014 |
| $\dot{e}_e^\dagger/e_e^\dagger$ | 0.036 | 0.001 | 0.008 | -0.013 |
| $\dot{e}_c^\dagger/e_c^\dagger$ | 0.004 | 0.004 | -0.013 | 0.001 |
| ln $REC/dt = \dot{e}_e^\dagger/e_e^\dagger - \dot{e}_c^\dagger/e_c^\dagger$ | 0.032 | -0.003 | 0.021 | -0.014 |

0.01). The parameter $\rho_3$ was set as 0.01 for ages younger than $a^\dagger$, but for ages greater than $a^\dagger$, $\rho_3$ was 0.013 in the "faster progress above $a^{\dagger}$" scenario; the opposite setting of $\rho_3$ is set for the "faster progress below $a^{\dagger}$" scenario, that is, $\rho_3 = 0.013$ for ages below $a^\dagger$ and 0.01 for ages above $a^\dagger$. In the case of the shifting mortality model, $\rho_3$ was set to be the same as 0.01, equal to $\rho_1$ and $\rho_2$, throughout adulthood (Table 3).

Fig 3 shows simulated life expectancy, threshold age, REC, $e^\dagger$, $e_c^\dagger$ and $e_e^\dagger$. For all the three scenarios, life expectancy and the threshold age maintain an upward trend (Fig 3A and 3B), with a gap of about 2 years between them. The "faster progress below $a^{\dagger}$" scenario yields the highest life expectancy because all adults younger than $a^\dagger$ can benefit from the faster progress of mortality decline. For "faster progress above $a^\dagger$," however, only mortality rates among older people can decline faster; thus, life expectancy is lower. Compared with the two scenarios with faster reduction of mortality at some ages, the case of the "shifting model," with baseline progress of mortality reductions, has the lowest life expectancy.

In the case of "faster progress below the threshold age," deaths are averted more at young ages than at old ages such that those averted deaths are postponed to, and concentrated in, a narrow interval, leading to a decrease in $e^\dagger$. In contrast, when old-age death rates decline more rapidly, older people survive longer, increasing the expansion component and, in turn, $e^\dagger$. It is somewhat surprising that the shifting mortality hypothesis does not lead to a constant $e^\dagger$, but rather a slow decline in $e^\dagger$.

As noted above, the REC can capture the link between the compression and expansion components and hence show different patterns than $e^\dagger$ (Fig 3D). For the scenarios of "faster progress below $a^{\dagger}$" and shifting mortality, the REC increases over time, indicating that the expansion component is important relative to the compression component. For these two cases, both $e_c^\dagger$ and $e_e^\dagger$ decrease, but $e_c^\dagger$ decreases faster than $e_e^\dagger$ (Fig 3E and 3F), thereby increasing the REC.

For the case of "faster progress above $e^\dagger$," the REC increases, peaks in year 100 (simulated) and then decreases. Because of the more rapid increase in survival at old age, $e_e^\dagger$ continues to increase linearly. However, $e_c^\dagger$ increases slowly until the year 100, when it begins to increase rapidly. Consequently, the REC shows a reverse-U shape over time. The decline in REC indicates a decrease in lifespan disparity because of the slow increase in the expansion component.

## Empirical REC

**Cohort trends in the REC.** Implicitly or explicitly, the compression or expansion of mortality is often interpreted from the perspective of a cohort. For example, because of infant

**Table 3. Parameters of the Siler's model under different scenarios.**

|  | Shifting mortality | Faster progress below $a^\dagger$ | Faster progress above $a^\dagger$ |
|---|---|---|---|
| Below $a^\dagger$ | $\rho_3 = 0.01$ | $\rho_3 = 0.013$ | $\rho_3 = 0.01$ |
| Above $a^\dagger$ | $\rho_3 = 0.01$ | $\rho_3 = 0.01$ | $\rho_3 = 0.013$ |

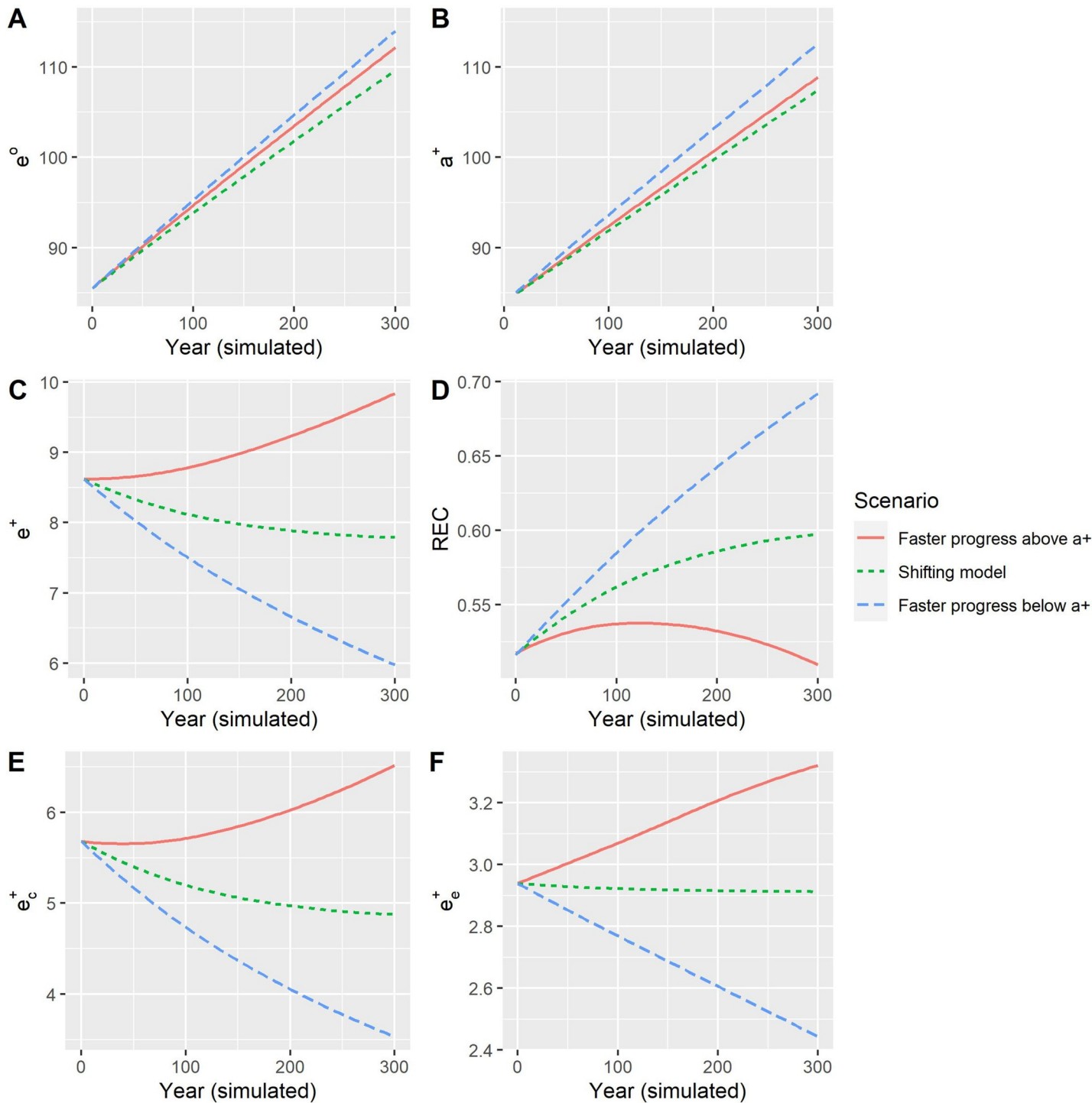

**Fig 3. Trajectories of $e^o$, $e^\dagger$, REC, $e^\dagger$, $e^\dagger_c$ and $e^\dagger_e$ in a Siler mortality change model with different rates of decline.**

mortality decline, some deaths that would have occurred are postponed and become increasingly concentrated in a narrow age range. A synthetic cohort is used in such a scenario, despite the fact that the data consist of period measures. We apply our new measure to real cohorts to further explore its properties.

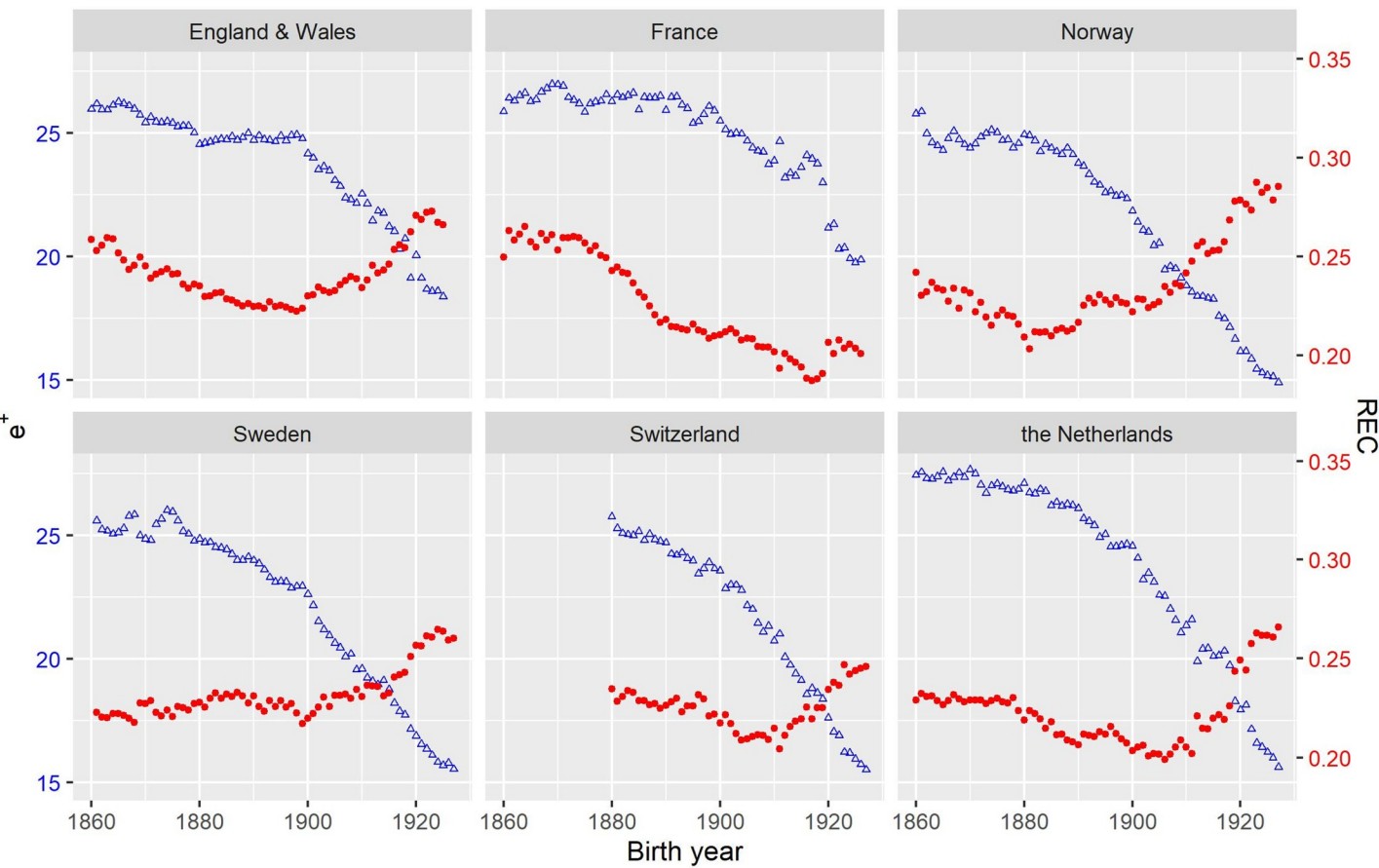

**Fig 4. Cohort life lost due to death (triangles in blue) and the REC (dots in red) of females from selected countries.**

Fig 4 shows the trends in $e^\dagger$ and the REC for real cohorts born between 1860 and 1928 based on data obtained from the Human Mortality Database [18]. $e^\dagger$ and the REC reveal different or even opposite trends in lifespan disparity. The successive cohorts born after 1860 experienced a significant decline in $e^\dagger$, although $e^\dagger$ was roughly constant for cohorts born before 1900. However, the REC shows somewhat different or even opposite trends. Except for Sweden, the REC slightly decreased and $e^\dagger$ declined until cohorts born in ca. 1900, consistently indicating mortality compression. However, the increase in the REC afterward reveals that the decline in $e^\dagger$ indicated an increasing importance of the expansion component. Sweden's REC remained nearly constant until the 1900 cohort. In short, the REC shows a similar trend to $e^\dagger$ for cohorts born before ca. 1900, but their trends began to diverge after ca. 1900. This divergence suggests that the cohorts born after 1900 had greatly benefited from improvements in old-age mortality that began in the 1960s.

**Period trends in the REC.** Although the cohort perspective provides a clear view of the evolution of compression or expansion, the time lag in a cohort study makes the information of little use to the public and policy makers, who are concerned with the current state of lifespan disparity.

As shown in Fig 5, the trend in the REC over time differed from that of $e^\dagger$. The period $e^\dagger$ rapidly declined for nearly 100 years from 1860, and the decline markedly slowed in the 1950s and continued to decline even more slowly thereafter. This decline largely stemmed from the shift in mortality reductions from early- and mid-age mortality to old-age mortality.

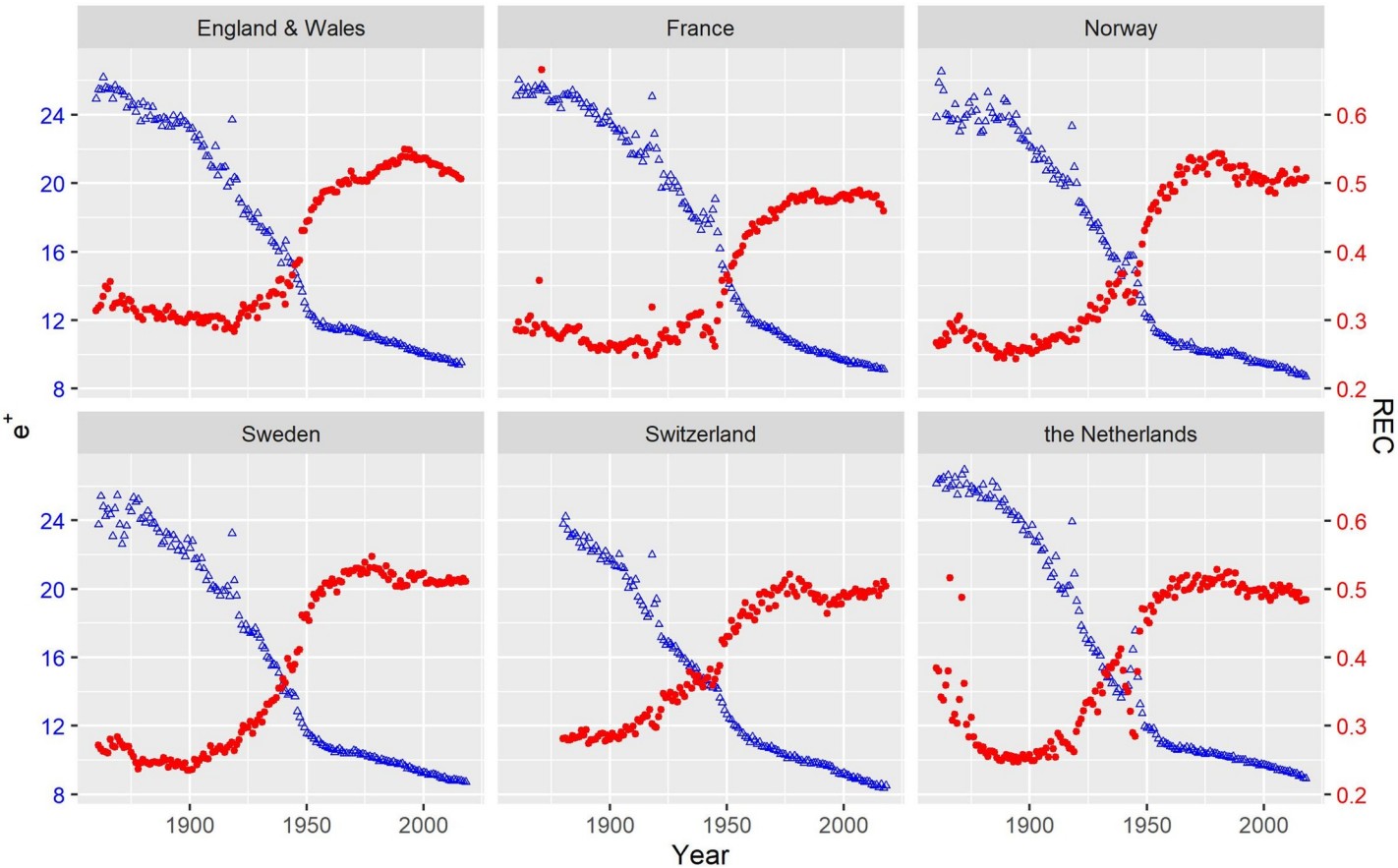

**Fig 5.** Period life lost due to death (triangles in blue) and the REC (dots in red), 1860–the latest data available of females from selected countries.

The REC remained constant until the 1900s–1930s and then rapidly decreased and peaked in the 1970s. Unlike other countries, England and Wales experienced an increase in the REC until the 1990s, followed by a gradual decline. The REC eventually reached 0.5; that is, the expansion component was approximately half of the compression component. Importantly, the constant REC suggests that the rise in the expansion could nearly offset the compression component, despite the continuous decline in $e^{\dagger}$. Note that the constant REC did not result from the shifting mortality, as the shifting mortality hypothesis predicts a strictly increasing pattern in the REC, as illustrated in the above simulations.

In addition, the trajectories of the REC fluctuated more strongly than those of $e^{\dagger}$. This difference largely stems from the fact that the REC can be affected by changes in both expansion and compression, while in $e^{\dagger}$, the two terms are only partially offset. Hence, the REC is more sensitive than $e^{\dagger}$ and can thus respond to more subtle changes.

## Discussion

Here, we developed a new measure, the REC, that can capture the essence of changes in lifespan disparity: changes in the relative importance of the expansion component relative to the compression component. Simulation and empirical results highlight the advantages of the REC for measuring lifespan disparity.

Unlike conventional aggregate measures, such as $e^{\dagger}$, wherein expansion and compression are only partially offset, the REC can capture detailed changes in expansion and compression,

including how changes in these components shape the trajectories of lifespan disparity. The advantage stems from the fact that any change in lifespan disparity represents a balance between expansion and compression. Thus, instead of simply interpreting a change in disparity as positive or negative, REC permits changes to be interpreted in terms of changes in the relative importance of expansion relative to compression.

Relative changes—not absolute changes—are key for comparing expansion and compression. Because of the curvature of related functions (e.g., $\rho(x)$ and $g(x)$), the compression component can be double or triple the expansion component. Moreover, the same pace of mortality decline can yield greater changes in the compression component than the expansion component. Hence, comparing absolute changes of the two terms can be misleading. Because the REC is a ratio that reflects the relative importance of one component to another, comparisons based on the REC avoid the problem of different scales.

Despite the advantages associated with the REC, we do not advocate that conventional measures of disparity could be abandoned. Instead, we suggest that both the REC and conventional measures should be used when appropriate. For example, in periods showing mortality transitions (e.g., around the 1950s), the relation between REC and $e^\dagger$ could be reversed. With $e^\dagger$ as a reference, the REC can provide a comprehensive perspective of lifespan disparity.

## Supporting information

**S1 Code. The code to reproduce the results and graphs presented in this section are publicly available through the repository in the link https://github.com/imoza/code.for.REC.** (TXT)

## Acknowledgments

We thank two reviewers (Ugofilippo Basellini and Viorela Diaconu, who identified themselves) and the Academic Editor Bernardo Lanza Queiroz for helpful comments and suggestions.

## Author Contributions

**Conceptualization:** Zhen Zhang, Qiang Li.

**Formal analysis:** Zhen Zhang, Qiang Li.

**Software:** Zhen Zhang.

**Writing – original draft:** Zhen Zhang, Qiang Li.

**Writing – review & editing:** Zhen Zhang, Qiang Li.

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
