## [Decision Letter · Decision Letter 0]

8 Oct 2020

PONE-D-20-25812

The ratio of expansion to compression: A new measure of lifespan disparity

PLOS ONE

Dear Dr. Li,

Thank you for submitting your manuscript to PLOS ONE. After careful consideration, we feel that it has merit but does not fully meet PLOS ONE’s publication criteria as it currently stands. Therefore, we invite you to submit a revised version of the manuscript that addresses the points raised during the review process.

This is a very good paper. It makes an important contribution and brings into the literature an relevant discussion. The topic is well presented and discussed. However, the paper has some shortcomings. The detailed comments from the reviewers are presented in the attached files. 

We look forward to receiving your revised manuscript.

Kind regards,

Bernardo Lanza Queiroz, Ph.D

Academic Editor

PLOS ONE

Journal Requirements:

Reviewers' comments:

Reviewer's Responses to Questions

**Comments to the Author**

1. Is the manuscript technically sound, and do the data support the conclusions?

Reviewer #1: Yes

Reviewer #2: Yes

2. Has the statistical analysis been performed appropriately and rigorously? 

Reviewer #1: Yes

Reviewer #2: Yes

3. Have the authors made all data underlying the findings in their manuscript fully available?

Reviewer #1: Yes

Reviewer #2: Yes

4. Is the manuscript presented in an intelligible fashion and written in standard English?

Reviewer #1: Yes

Reviewer #2: Yes

5. Review Comments to the Author

Reviewer #1: This is an interesting study and the authors have provided insight into the dynamics responsible for changes in lifespan disparity. They have made a great effort in documenting not only period trends in the proposed disparity indicator, but also cohort trends. This paper will contribute to our better understanding of how mortality changes at younger and older ages affected lifespan disparity.

The paper is generally well written and structured. However, in my opinion the paper has some shortcomings in regards to the presentation and explanation of some of the results. My detailed comments and suggestions can be found in the attached document.

Reviewer #2: I have written the review and comments to the authors in a separate document due to math symbols and comments in a formula, which would not appear in this form. Please refer to the comments to authors document attached.

6. PLOS authors have the option to publish the peer review history of their article (what does this mean?). If published, this will include your full peer review and any attached files.

Reviewer #1: No

Reviewer #2: **Yes: **Ugofilippo Basellini

---

## [Author Response · Author response to Decision Letter 0]

16 Oct 2020

We thank two reviewers and the Academic Editor for helpful comments and suggestions. Please see "Response to reviewers" for detailed response.

---

## [Decision Letter · Decision Letter 1]

18 Nov 2020

PONE-D-20-25812R1

The ratio of expansion to compression: A new measure of lifespan disparity

PLOS ONE

Dear Dr. Li,

Thank you for submitting your manuscript to PLOS ONE. After careful consideration, we feel that it has merit but does not fully meet PLOS ONE’s publication criteria as it currently stands. Therefore, we invite you to submit a revised version of the manuscript that addresses the points raised during the review process.

We appreciate the careful revision and the response to all comments and suggestions. Based on the reviewers comments and my own reading, we believe there are a few adjustments to be work in the paper before final acceptance.  There are just a few points to clarify the discussion and a few issues on language editing. We all agree this is an important and relevant contribution. 

We look forward to receiving your revised manuscript.

Kind regards,

Bernardo Lanza Queiroz, Ph.D

Academic Editor

PLOS ONE

Reviewers' comments:

Reviewer's Responses to Questions

**Comments to the Author**

1. If the authors have adequately addressed your comments raised in a previous round of review and you feel that this manuscript is now acceptable for publication, you may indicate that here to bypass the “Comments to the Author” section, enter your conflict of interest statement in the “Confidential to Editor” section, and submit your "Accept" recommendation.

Reviewer #1: (No Response)

Reviewer #2: (No Response)

2. Is the manuscript technically sound, and do the data support the conclusions?

Reviewer #1: Yes

Reviewer #2: Yes

3. Has the statistical analysis been performed appropriately and rigorously? 

Reviewer #1: Yes

Reviewer #2: Yes

4. Have the authors made all data underlying the findings in their manuscript fully available?

Reviewer #1: Yes

Reviewer #2: Yes

5. Is the manuscript presented in an intelligible fashion and written in standard English?

Reviewer #1: Yes

Reviewer #2: Yes

6. Review Comments to the Author

Reviewer #1: Dear authors,

Thank you for adequately addressing my previous comments and suggestions on your manuscript. While reading the revised version of your manuscript, I found that the newly added sections need some improvement.

Comment 1:

I find that the explanations of the different scenarios under which REC increases/decreases are not clear enough. My understanding from the explanations provided later on for Fig. 1, is that REC and its components depend on the mortality changes before and after the threshold age. I think it is important to mention this in the scenarios. For instance, when mortality at younger ages increases and mortality at older ages declines (?), increases (?), this is what happens to the REC. More precisely:

Line 146: “The REC can increase in two cases. The first case is a rise in young-age mortality […]”. In this first case, changes that occur in mortality at older ages do not matter?

Line 150: “In the second case, when old-age mortality has a faster decline […]”. Faster decline than what? I assume than at younger ages.

Also, the REC can increase in two cases. Can it not increase? Is there any doubt on the fact that it increases? Does it depend on something else? For me, the “can increase” formulation suggests that there is a possibility that under some condition it will not increase.

Line 155: I will not mention in this paragraph the decline in REC given that you explain at line 168 the scenarios under which REC declines.

Line 171: In the second case […]. What are the mortality conditions under this case?

Comment 2:

Line 60-62: "In a country with increasing younger adult mortality (because of wars or political distress), the compression component can rise and increase lifespan disparity".

Line 146: "The first case is a rise in young-age mortality, with a corresponding decrease in life expectancy and threshold age. The compression term may decrease because the interval [0, †] decreases".

Following an increase in mortality at younger ages, the compression component can increase (lines 60-62), but may decreases (lines 146). Which one is it?

Comment 3:

Figure 2: legend of colors doesn’t show

Reviewer #2: I would like to thank the authors for addressing all the comments and suggestions that I raised in the first round of reviews. I believe that the manuscript has improved and gained in clarity with respect to the previous draft. Also, I thank the authors for acknowledging my name in the Acknowledgment section, I really appreciated it and I am glad that my comments helped them to improve the manuscript.

I have three final minor comments for the authors:

- line 148: change "morality" with "mortality"

- line 190: "The disparity will rise when \\dot{e}^{\\dagger}_e is greater than \\dot{e}^{\\dagger}_c and viceversa". There is a minus missing in \\dot{e}^{\\dagger}_c

- Figure 2: please add back the legend of the years for easier readability

7. PLOS authors have the option to publish the peer review history of their article (what does this mean?). If published, this will include your full peer review and any attached files.

Reviewer #1: **Yes: **Viorela Diaconu

Reviewer #2: **Yes: **Ugofilippo Basellini

---

## [Author Response · Author response to Decision Letter 1]

19 Nov 2020

We thank two reviewers for helpful comments and suggestions.

---

## [Editor Report · Decision Letter 2]

23 Nov 2020

The ratio of expansion to compression: A new measure of lifespan disparity

PONE-D-20-25812R2

Dear Dr. Li,

We’re pleased to inform you that your manuscript has been judged scientifically suitable for publication and will be formally accepted for publication once it meets all outstanding technical requirements.

Kind regards,

Bernardo Lanza Queiroz, Ph.D

Academic Editor

PLOS ONE
---

## [Editor Report · Acceptance letter]

3 Dec 2020

PONE-D-20-25812R2 

The ratio of expansion to compression: A new measure of lifespan disparity 

Dear Dr. Li:

I'm pleased to inform you that your manuscript has been deemed suitable for publication in PLOS ONE. Congratulations! Your manuscript is now with our production department. 

Kind regards, 

on behalf of

Dr. Bernardo Lanza Queiroz 

Academic Editor

PLOS ONE